# The Effects of Thermocycling on the Physical Properties and Biocompatibilities of Various CAD/CAM Restorative Materials

**DOI:** 10.3390/pharmaceutics15082122

**Published:** 2023-08-10

**Authors:** Se-Young Kim, Han-Jin Bae, Hae-Hyoung Lee, Jong-Hyuk Lee, Yu-Jin Kim, Yu-Sung Choi, Jung-Hwan Lee, Soo-Yeon Shin

**Affiliations:** 1Department of Prosthodontics, College of Dentistry, Dankook University, 119 Dandae-ro, Cheonan 31116, Republic of Korea; tpdudsla2741@gmail.com (S.-Y.K.); hyuk928@dankook.ac.kr (J.-H.L.); 2Institute of Tissue Regeneration Engineering (ITREN), Dankook University, 119 Dandae-ro, Cheonan 31116, Republic of Korea; bhj137@naver.com (H.-J.B.); haelee@dku.edu (H.-H.L.); 3Department of Nanobiomedical Science & BK21 PLUS NBM Global Research Center for Regenerative Medicine, Dankook University, 119 Dandae-ro, Cheonan 31116, Republic of Korea; 4Cell & Matter Institute, Dankook University, 119 Dandae-ro, Cheonan 31116, Republic of Korea; 5Department of Biomaterials Science, College of Dentistry, Dankook University, 119 Dandae-ro, Cheonan 31116, Republic of Korea; yujin10316426@gmail.com; 6UCL Eastman-Korea Dental Medicine Innovation Centre, Dankook University, 119 Dandae-ro, Cheonan 31116, Republic of Korea; 7Mechanobiology Dental Medicine Research Center, Dankook University, Cheonan 31116, Republic of Korea; 8Department of Regenerative Dental Medicine, School of Dentistry, Dankook University, Cheonan 31116, Republic of Korea

**Keywords:** hybrid CAD/CAM restorations, thermocycling, physical properties, surface properties, biocompatibility

## Abstract

The purpose of this study is to evaluate the changes in physical properties and biocompatibilities caused by thermocycling of CAD/CAM restorative materials (lithium disilicate, zirconia reinforced lithium silicate, polymer-infiltrated ceramic network, resin nanoceramic, highly translucent zirconia). A total of 225 specimens were prepared (12.0 × 10.0 × 1.5 mm) and divided into three groups subjected to water storage at 37 °C for 24 h (control group), 10,000 cycles in distilled water at 5–55 °C (first aged group), and 22,000 cycles in distilled water at 5–55 °C (second aged group) [(*n*= 15, each]). The nanoindentation hardness and Young’s modulus (nanoindenter), surface roughness (atomic force microscopy (AFM)), surface texture (scanning electron microscopy (FE-SEM)), elemental concentration (energy dispersive spectroscopy (EDS)) and contact angle were evaluated. The morphology, proliferation and adhesion of cultured human gingival fibroblasts (HGFs) were analyzed. The data were analyzed using one-way ANOVA and Tukey’s test (*p* < 0.05). The results showed that the nanoindentation hardness and Young’s modulus were decreased after thermocycling aging. Cell viability and proliferation of the material decreased with aging except for the highly translucent zirconia. Zirconia-reinforced lithium silicate exhibited significantly lower cell viability compared to other materials. The surface roughnesses of all groups increased with aging. Cell viability and Cell adhesion were influenced by various factors, including the surface chemical composition, hydrophilicity, surface roughness, and topography.

## 1. Introduction

Ceramics exhibit high wear resistance, aesthetics, and excellent color stability, as well as a high degree of biocompatibility [1]. However, they also accelerate the wear cause by opposing dentition and they are prone to fracture due to their brittleness [2]. In contrast, resin composites have the appropriate esthetics, easy milling processes, easy repairability in the oral cavity, and lower wear for opposing dentition compared to ceramics [1,3]. However, the resin composites have low color stabilities and reduced surface glosses after wear [4,5].

To address the drawbacks of these two materials, there has been a demand for new CAD/CAM materials that combine their strengths. As a result, hybrid materials have been introduced, and they can be classified into the following categories based on their manufacturing methods, compositions, and microstructures: polymer-infiltrated ceramic network (PICN), resin nanoceramic (RNC), and zirconia-reinforced lithium silicate (ZLS).

PICN consists of mesh-like skeletons of porous feldspar-based ceramic materials, with both ceramic (86 wt.%) and polymer (14 wt.%) properties due to the interpenetration of the two materials achieved by polymer injection [6]. On the other hand, RNC is composed of nanoscale silica and zirconia inorganic filler particles embedded in an organic polymer matrix, resembling a composite resin. Similar to PICN, RNC exhibits a combination of ceramic and polymer properties [7]. ZLS is composed of a silica-based glass matrix with 10% dissolved zirconia. In contrast to lithium disilicate blocks (IPS e.max CAD, Ivoclar Vivadent, Schaan, Liechtenstein), which necessitate a crystallization process after machining, ZLS only require grinding or sintering when high strength is needed [8].

These materials combine the advantages of different materials and offer improved esthetics resembling natural teeth and superior resistance to discoloration. Compared to ceramics, the processabilities are better, so the milling times are shortened, and the burs have longer lifespans. Additionally, they are reported to have easier repairability in the oral environment than traditional ceramics [9,10,11,12,13,14].

While these ceramic materials offer exceptional aesthetics, they have higher silica contents and lower strengths compared to materials based on dense zirconia polycrystals. As a result, they may be less suitable for situations that require withstanding high stress concentrations [15].

A few companies have recently introduced a new type of zirconia material called high-translucent zirconia, which has a smooth surface and exhibits natural color transitions. High-translucent zirconia typically contains a lower concentration of yttria than 3Y-TZP zirconia. The reduced yttria content in the high-translucent zirconia provides the increased translucency [16]. Generally, all the aforementioned materials have been recommended for use in all-ceramic monolithic restorations. They are conveniently available in prefabricated blocks specifically designed for CAD/CAM systems.

Thermocycling is an effective way to expedite simulated aging of the samples. It is useful because the temperature of the oral environment can be reproduced to estimate the clinical performance. Variations in the thermal and fatigue resistances of these materials provide information about their clinical failures. The temperature changes that occur when drinking and eating cold and hot substances lead to contraction and expansion of the restorative materials. As a result of these changes, mechanical stresses and cracks occur [17]. Thus, the thermocycling method can affect the longevity of the restoration, and using it can simulate the behavior of the ceramic material in the oral environment [18].

In terms of biocompatibility, it is important to examine the potential negative impacts on human oral soft tissues. HGFs (human gingival fibroblasts) play an essential role in maintaining the integrity of the connective tissues during the process of oral wound healing and regeneration [19]. The CAD/CAM materials utilized for coronal restorations, whether with natural teeth or implants, will be in direct contact with oral soft tissues, including the keratinized marginal gingiva [20,21]. Studies on the biocompatibilities of existing CAD/CAM materials have only been conducted with individual materials, and there are few comparative studies of cell viability and proliferation for various types of hybrid CAD/CAM ceramics and zirconia. Thus, evaluations of HGF cells for various CAD/CAM hybrid restorations are needed after inducing artificial aging processes that simulate natural aging [22,23,24,25]. Additionally, when evaluating cell adhesion, the surface roughness, chemical compositions and hydrophilicities of the materials should be considered [26,27]. According to several literature sources, numerous surface properties are believed to influence the attachment and proliferation of human gingival fibroblasts (HGFs). The effect of surface roughness on attachment and proliferation is a subject of controversy, with certain studies suggesting that smooth surfaces are more favorable, while others indicate the opposite [28,29]. Therefore, this in vitro study investigated the physical and surface properties as well as the biocompatibilities of materials under thermal cycling aging, which is one of the aging processes. The null hypothesis presented in this study is that the physical properties of various CAD/CAM restorative materials do not differ before and after aging, the biocompatibility of various CAD/CAM restorative materials does not differ before and after aging, and the surface properties of these restoration materials are no difference before and after aging, and that the surface characteristics are not associated with cell adhesion.

## 2. Materials and Methods

### 2.1. Sample Preparation

Five CAD/CAM materials were prepared, including lithium disilicate (M, [IPS e.max CAD; Ivoclar vivadent AG, Schaan, Liechtenstein]) used as a glass matrix ceramic, a zirconia-reinforced lithium silicate glass ceramic (C, [CeltraDuo; Densply Sirona, Charlotte, NC, USA]), a polymer-infiltrated ceramic network (E, [Vita Enamic; Vita Zahnfabrik, Bad Säckingen, Germany]), a resin nanoceramic (S, [Cerasmart; GC DentalProducts, Tokyo, Japan]) hybrid CAD/CAM material, and high translucency zirconia (Z, [Lava^TM^ Plus Zirconia; 3 M/ESPE, Neuss, Germany]). The types and other information are shown in Table 1.

Every block was fabricated as a plate with dimensions of 12.0 mm width × 10.0 mm length × 1.5 mm height (Figure 1a). Among them, the IPS e.max CAD and Lava^TM^ Plus zirconia were produced in their final forms after additional sintering with Programat EP 5000 (Ivoclar vibadent, Schaan, Lichtenstein) and Camelon (CM Therm, Neo Biotech, Seoul, Republic of Korea) furnaces, respectively, according to the manufacturers’ instructions.

A total of 225 specimens were fabricated, and they were cleaned in distilled water with an ultrasonic cleaner (SD-120H, Mujigae Co., Seoul, Republic of Korea) for 10 min and then air-dried at room temperature for 24 h. The dried specimens were divided into 25 groups (N = 25 per group) based on the following criteria: a control group without aging, a first aging group subjected to 10,000 cycles of thermal cycling, and a second aging group subjected to 22,000 cycles of thermal cycling (Table 2).

### 2.2. Aging Procedure

Five samples of each material were randomly selected. The first aged group was subjected to thermal cycling (TW-D813, Taewon tech, Bucheon-si, Republic of Korea) (Figure 1b) in distilled water at 5–55 °C for 10,000 cycles (5 °C for 30 s; 55 °C for 30 s; rest period; 10 s), which corresponded to approximately 1 year in the clinical setting. Another group (second aged) underwent thermal cycling for 22,000 cycles under the same conditions, which corresponded to approximately 3 years in a clinical environment [30,31]. The control group was stored in distilled water at 37 °C for 24 h.

### 2.3. Measurements of Physical Properties

#### Nanoindentation Hardness and Young’s Modulus (Elastic Modulus)

The nanoindentation hardness and Young’s modulus (Elastic modulus) were measured with a Nanoindenter (Nanoindenter^®^ XP MTS, Chicago, IL, USA). The Berkovich diamond tip roundness was 40 nm, the essential load was 45 mN, the strain rate was 0.05%/s, the maximum penetration depth was 2 µm, and each specimen was measured 3 times.

### 2.4. Analyses of Surface Properties

#### 2.4.1. Energy Dispersive X-ray Spectroscopy (EDS)

Energy dispersive X-ray spectroscopy (SU8230; Hitachi High Technologies, Tokyo, Japan) with a measurement voltage of 15 kV was used to observe differences in the surface components before and after aging.

#### 2.4.2. Field-Emission Scanning Electron Microscopy (FE-SEM)

Energy dispersive X-ray spectroscopy (SU8230; Hitachi High Technologies, Tokyo, Japan) with a measurement voltage of 15 kV at 1000 magnifications was used to observe differences in the surface components before and after aging.

#### 2.4.3. Atomic Force Microscopy (AFM)

To analyze the surface roughnesses (R_a_, R_q_), atomic force microscopy (XE-100;Park Systems, Suwon, Republic of Korea) was used for 3 measurements per specimen. The scan sizes were 10 × 10 µm^2^, the resolution was 256 × 256 pixels, and the scan rate was 0.8 Hz. And the non-contact mode Kelvin probe force microscopy equipped with a Pt/Cr-coated silicon tip (tip radius < 25 nm, force constant 3 N m^−1^, and a resonance frequency of 75 kHz) was employed to measure the surface charge potentials.

#### 2.4.4. Contact Angle Measurements

The surface wettabilities of the samples were characterized by water contact angle measurements with a contact angle analyzer (Phoenix 300; SEO, Suwon, Republic of Korea). The sessile drop method was used for contact angle measurements. Images of the drops were captured with a video camera as soon as they landed on the surfaces. In each case, three water droplets were measured for each test specimen. ImageJ software (National Institutes of Health, Bethesda, MD, USA) was used to analyze the images and measure the contact angles. The average angle of the three droplets was obtained.

### 2.5. Analysis of Biological Properties

#### 2.5.1. Cell Culture of Human Gingival Fibroblasts

Human gingival fibroblasts (HGFs) were isolated from gingival fragments obtained by surgical tooth extraction from 23-year-old female patients (IRB No. DKUDH IRB 2023-06-001) at the Oral and Maxillofacial Surgery Department, Dental Hospital, Dankook University. All participants provided written informed consent. After the gingival fragments were collected, the fresh tissue was preserved in 5 mL of Hank’s balanced salt solution (HBSS; Welgene, Daegu, Republic of Korea) and 1% penicillin/streptomycin (PS, Gibco^TM^, Thermo Fisher Scientific, Waltham, MA, USA). The tissue was chopped and submerged in a 2 mg/mL collagenase type I solution (Worthington Biochemical, 1icpa water bath) for 1 h at 37 °C, and then the gingival epithelial layer was separated and cut into approximately 1 × 1 mm size and placed in culture dishes (Falcon^®^, Corning, NewYork, NY, USA). The tissue was cultured in 2 mL of growth medium (αMEM, MEM Alpha Modification, with Ribo- and deoxyribonucleosides, with L-glutamine, HyClone, Logan, UT, USA) containing 10% fetal bovine serum (FBS^®^, Corning), 2 mM L-glutamine (Gibco™ Thermo Fisher Scientific, Waltham, MA, USA), 100 mM non-essential amino acids solution (100×, Gibco™ Thermo Fisher Scientific), 55 µM 2-mercaptoethanol (1000×, Gibco™ Thermo Fisher Scientific), and growth medium with 1% PS. The HGF cells were cultured in αMEM supplemented with 10% FBS, 1% MEM nonessential amino acid solution (MEM NEAA; 11140050, Gibco™ Thermo Fisher Scientific), 1% GlutaMAX™ supplement (3505006, Gibco™ Thermo Fisher Scientific), 1% penicillin–streptomycin (15140-163, Gibco™ Thermo Fisher Scientific) and 0.2% 2-mercaptoethanol (21985023, Gibco™ Thermo Fisher Scientific) at 37 °C in a 5% CO_2_ incubator. The dental CAD/CAM blocks were placed on nontreated plates and the plates were seeded, and the experiments were performed at 4 and 24 h after seeding. The cells were then separated with 0.25% trypsin-EDTA (#25200114, Gibco™, Grand Island, NY, USA) and passaged when the cell confluence reached 80–90%, and passages 5–10 were used for the following experiment.

#### 2.5.2. Cell Viability Tests with the LIVE/DEAD Assay & Cell Counting Kit-8 (CCK-8) Assay

##### LIVE/DEAD Assay

Samples were placed in 12-well non treated plates (#32012, SPL Life Science Co., Ltd., Gyeonggi, Republic of Korea), the HGFs were seeded on samples at a density of 5.0 × 10^4^ cell per well and the experiment was performed three times. After 24 h, the samples were rinsed with PBS and then incubated in the Live/Dead solution (2 μM Calcein-AM and 4 µM EthD-1, InvitrogenTM, Carlsbad, CA, USA) for 30 min at 37 °C. Then, a fluorescence microscope (Olympus IX71, Olympus, Tokyo, Japan) was used to visualize the live cells exhibiting green fluorescence, and dead cells exhibited red fluorescence.

##### CCK-8 Assay

The Cell Counting Kit-8 (CCK-8, CK04-20, Dojindo, Japan) assay was used to assess the activity of the mitochondrial dehydrogenase in the living cells, which is a marker of cell viability. First, three samples per group were placed in 12-well untreated plates, seeded with cells at a density of 5.0 × 10^4^ per well, and incubated for 24 h. After 24 h of incubation, the samples were washed with PBS and then incubated for 2 h at 37 °C in 1 mL of culture medium with 100 μL of CCK solution added to each well. The optical density was measured at 450 nm with a microplate reader (Scientific Varioskan LUX Multimode Microplate Reader, Thermo Fisher Scientific, Waltham, MA, USA).

#### 2.5.3. Immunocytochemistry of Cell Spreading and Focal Adhesion

The HGFs were seeded in 24-well nontreated plates (#32024, SPL Life Science Co., Ltd., Gyeonggi, Republic of Korea) at a density of 2 × 10^4^ cells per well and cultured for 4 and 24 h. After washing with Hanks’ balanced salt solution (LB 003-02 HBSS; Welgene, Daegu, Republic of Korea), the cells were fixed in 4% paraformaldehyde (PFA; T&I, BFA-9020, Kyunggi-Do, Republic of Korea) for 15 min at RT. After fixation, the HGFs were permeabilized with 0.2% Triton X-100 in PBS (Sigma Aldrich, Saint Luis, MO, USA, #9002-93-1) for 10 min and blocked with a 1% bovine serum albumin fraction (BSA; SM-BOV-100, Geneall, Seoul, Republic of Korea) for 1 h.

Then, the cells were stained with Alexa Fluor™ 546-conjugated phalloidin (#A22283, Invitrogen™, Carlsbad, CA, USA) to visualize the filamentous actin (F-actin) and stained with SYTOX™ Green nucleic acid stain (S7020, Invitrogen™, Carlsbad, CA, USA) for 10 min at RT to visualize the cell nuclei.

To investigate the cellular focal adhesion, the focal adhesion protein as Talin 1 and Vinculin were used for immunofluorescence labeling. The cells incubated with mouse anti-talin 1 monoclonal primary antibody (#sc-81805, SantaCruz Biotechnology, CA, USA) and rabbit anti-vinculin primary antibody (#ab129002, Abcam, Cambridge, UK) overnight at 4 °C and stained with anti-rabbit secondary antibodies conjugated to tetraethyl rhodamine isothiocyanate (TRITC; 715-095-150, Jackson ImmunoResearch,, PA, USA) or fluorescein isocyanate (FITC; 711-025-152, Jackson ImmunoResearch).

Finally, a fluorescence microscope (Olympus IX71, Olympus, Tokyo, Japan) was used to collect the fluorescence images. The images were quantified with ImageJ software to determine the F-actin intensity, cell area, Talin 1 intensity and Vinculin intensity.

## 3. Results

### 3.1. Physical Properties

#### 3.1.1. Nanoindentation Hardness

The means and standard deviations of the nanoindentation hardness before and after aging are presented for all materials Figure 2a, Appendix A. In all groups except for zirconia, the nanoindentation hardness of both of the aged groups was significantly lower than that of the control group. The nanoindentation hardness was found to be the lowest in the Group S, while the Group Z exhibited the highest values among all the groups before and after aging (*p* < 0.05). The values of Groups M, C, E, and S exhibited significant decreases after the 2nd aging (*p* < 0.05).

#### 3.1.2. Young’s Modulus (Elastic Modulus)

The mean and standard deviations for the Young’s moduli determined for each material before and after aging are presented in Figure 2b, Appendix A. Group S exhibited the lowest value among all the tested materials before and after aging (*p* < 0.05). There were no significant differences in Young’s moduli of Group M and Group C (*p* > 0.05), while significant differences were observed for all other materials both before and after aging. The values of all material groups exhibited significant decreases after aging (*p* < 0.05). Group M showed a decrease between the 1st age group and the 2nd aged group.

### 3.2. Surface Properties

#### 3.2.1. Energy Dispersive X-ray Spectroscopy (EDS)

In comparing the surface compositions of all materials before and after aging, there were several elements that showed statistically significant differences between the materials, as described in detail in Table 3.

Group M was mainly composed of C, O, Al, Si, P, and K, and the carbon and potassium components showed significant decreases as the aging process progressed. In Group C, while the carbon (C) and oxygen (O) levels increased in the 2nd aged group compared to the 1st aged group, the zirconium (Zr) content decreased significantly. Group E was mainly composed of C, O, Al, Si, Na, and C was the base of the polymer. In the 2nd aged group, the carbon (C) and oxygen (O) contents showed significant increases compared to the control group. Group S was mainly composed of C, O, Al, Si, and Ba. There was little difference in the elemental composition after aging. Group Z, which is a type of zirconia, showed an increase in the carbon (C) content as the aging process progressed, while the zirconium (Zr) content decreased in contrast.

#### 3.2.2. Field-Emission Scanning Electron Microscopy (FE-SEM)

FE-SEM images showed that the topographic patterns differed among all groups (Figure 3). The surface of the control group was relatively smooth, whereas the aged group showed micro irregularities, porosity, and defects. In Group M, rectangular shaped lithium disilicate particles were well observed, but some irregular shape was observed after aging. In Group E, plate-like ceramic supports were well observed in all groups and remained after aging, Group S exhibited uniformly sized ceramic fillers dispersed in all groups, and no significant changes were observed even after aging. And Polycrystalline particles were observed in Group Z, and changes in crystal morphology were observed after aging.

#### 3.2.3. Atomic Force Microscopy (AFM)

Representative AFM images for all groups are shown in Figure 4. In the control group of all materials, the grains exhibited a smooth surface with a dense network of interlocking acicular branches, and the spaces between them were filled with a compact amorphous phase, such as the glassy matrix or ground mass. After aging, the surface texture showed localized surface uplifts on the grains. Surface roughness and microcracks primarily progressed along the grain boundaries.

The mean and standard deviation for the surface roughness (R_a_ and R_q_) measured before and after aging for each group, as well as the statistical analyses, are presented in Figure 5a,b and Appendix A. Both before and after aging, Group S showed the highest Ra and Rq values, while Group Z showed the lowest Ra and Rq values. In all materials, the Ra and Rq values increased relative to that of the control group as aging progressed. In Group M, Rq and Sq showed significant increases in the 2nd aged group compared to the control.

#### 3.2.4. Contact Angle Measurements

The means, standard deviations, and statistical analyses of the contact angle strengths are presented in Figure 5c,d and Appendix A. In all groups, the contact angles were significantly lower for the 2nd aged group than for the control group (*p* < 0.05). Group Z exhibited the highest contact angles before and after aging, followed by Groups C, S, E, and M. In both Groups C and Z, the contact angles decreased significantly as the aging time increased (*p* < 0.05). Groups E and S showed similar contact angle values before and after aging.

### 3.3. Cell Properties

The following are the experimental results of the biocompatibility analysis of various CAD/CAM materials. The HGF cells were cultured on specimens to analyze cell viability through Cell Counting Kit-8 (CCK-8, Dojindo Laboratories, Kumamoto, Japan) assay and LIVE/DEAD assay, and cell spreading and focal adhesion through immunofluorescence staining (Figure 6a).

#### 3.3.1. Cell Viability

Figure 6 shows the cell viability results after incubation of HGFs in five types of materials for 24 h. The cell viability was evaluated by measuring the cell viability of HGFs upon aging by thermocycling for each material using the CCK-8 assay, and then re-evaluated by fluorescence microscopy images using the LIVE/DEAD assay.

The group M and the group C showed no significant difference in cell viability between the control and 1st aged groups, but there was a tendency for HGF cell viability to decrease in the 2nd aged group (Figure 6b). The group E and group S also showed a decrease in HGFs viability in all aged groups compared to the control. However, the group Z increased the viability of HGFs in the aged group compared to the control group.

To confirm cell viability in culture conditions, LIVE/DEAD staining was performed and confirmed by fluorescence microscopy images (Figure 6c). Live cells are stained green and dead cells are stained red, with dead cells marked with white arrows. The percentage of dead cells in all groups was low, which means that most of the cells were able to survive on the surface, which indicates that all samples exhibited good biocompatibility for HGFs. Also, as shown in the image, cell viability decreased with aging, except for the group Z.

#### 3.3.2. Immunocytochemistry of Cell Spreading and Focal Adhesion

Cell size and spreading were assessed F-actin cytoskeleton stained with phalloidin (red) and nuclear stained with SYTOX™ (green) followed by fluorescence microscopy images (Figure 7a). As shown in the images, the cells cultured in the control and aged groups of each material look different in cell morphology, either wider or thinner, but all are well spread. To assess the size and spread of the cells, the cell area was quantified using Image J software. In all materials, there was no significant difference in cell area with aging. We also quantified and analyzed the intensity of actin, which is important for cell structure and movement. All material groups except the group E showed lower f-actin intensity in all aged groups compared to the control group. This suggests that the ability of cells to adhere or growth may be reduced in the aged groups. On the other hand, the group E showed higher f-actin intensity in the aged group compared to the control group. This indicates that the cells in the aged group are able to adhere and growth more effectively to the material surface.

To further investigate the cell adhesion of HGF to the materials, we analyzed the focal adhesion markers Talin1 and Vinculin by staining after 4 h of HGFs incubation on the five materials (Figure 7b). The quantification of Talin1 and Vinculin showed that the intensity of focal adhesion related proteins decreased in the aged groups of the group M, group C and group Z compared to the control group. This suggests that cell adhesion and cell motility may have decreased in the aged groups. However, in the group C and group E, the intensity of focal adhesion related proteins decreased in the aged group compared to the control group. This suggests that the formation or stability of focal adhesions may have increased with aging.

## 4. Discussion

Dental restorative materials undergo chemical, biological, physical, and thermal changes due to the complex oral environment [32]. Changes in oral temperature can lead to decreases in the strengths of restorations and accelerate the progression of cracks [33,34]. In previous studies, various accelerated aging procedures have been applied to conventional CAD/CAM restorative materials, and thermal cycling is one of the most commonly used methods for aging dental restorative materials [35]. Therefore, in this study, the physical properties, surface properties, and biological properties of various hybrid CAD/CAM restorations before and after thermal cycling aging were compared.

The data suggested that the physical properties of the tested materials exhibited gradual declines after thermocycling; thus, the first null hypothesis suggesting that the physical properties of the different CAD/CAM restorations would not be affected by thermal cycling aging is rejected. All CAD/CAM ceramic materials except for high translucence zirconia (Group Z) showed statistically significant decreases in hardness and elastic moduli with increasing thermocycling. This occurred because thermocycling involved cyclic temperature changes, which induced thermal stresses within the material. These thermal stresses can lead to microstructural changes, such as grain boundary cracking or phase transformations, and these weaken the material and result in decreased hardness and elastic modulus. Additionally, the second null hypothesis suggesting that the material type would not affect the properties of different CAD/CAM restorations is also rejected. The nanoindentation hardness and Young’s modulus values were highest for the Lava plus zirconia (Group Z), followed by IPS e.max CAD (Group M) and then the hybrid CAD/CAM materials. Zirconia was less affected by thermocycling due to its low thermal expansion coefficient, structural stability, and high crystal structure density. These characteristics allowed the zirconia to maintain its mechanical properties more effectively during thermocycling conditions.

IPS e.max CAD (Group M) is a glass-ceramic material, which means it contained both glass and crystalline phases. The presence of crystalline particles in the material contributed to the higher hardness and elastic modulus values compared to the purely resin-based hybrid CAD/CAM ceramics such as the resin nanoceramic (RNC) and polymer-infiltrated ceramic network (PICN). Among the resin-based hybrid CAD/CAM restorations, Cerasmart (Group S) showed significantly lower values even after thermal cycling, followed by Vita Enamic (Group E). This was consistent with previous studies. Alamoush et al. [36] evaluated the physical properties of various CAD/CAM ceramics and found that the specimens fabricated from RNC recorded the lowest nanoindentation hardness values and elastic moduli, followed by those from PICN. This was explained by the fact that PICN is composed of two continuous interpenetrating networks: a ceramic network and a polymer network, which differentiates it from other resin composite CAD/CAM blocks that are typically made up of a resin matrix with dispersed ceramic fillers.

The EDS and SEM analyses revealed significant alterations in the surface microstructures and compositions of the tested materials after thermal cycling aging. The initially homogeneous structures of the materials were found to deteriorate, and they exhibited noticeable surface uplifts, microcracks, and irregular defects and slight damage after aging. These changes were attributed to reductions in the particle sizes caused by aging, which in turn affected the elemental compositions of the materials. Vita Enamic (Group E) and Cerasmart (Group S) exhibited clear alterations in their surface microstructures, whereas IPS e.max CAD (Group M) and Celtra Duo (Group C) exhibited relatively slight alterations. As a result, thermal cycling had a greater impact on polymer-based materials.

The surface roughnesses had increased for all groups after the aging process. This was consistent with a previous study showing that simulated accelerated aging led to increases in the surface roughnesses of ceramic materials, which was attributed to the crystalline components, and the difference in the dissolution rates may have led to the increased surface roughnesses [37]. Among the materials, Cerasmart (Group S) showed the highest Ra value. The results of this study were consistent with previous findings indicating that when water is absorbed by the resin matrix, it causes expansion and plasticization of the resin, which lead to hydrolysis of the silane coupling agent and eventual loss of the surface fillers. Additionally, the absorbed water can diffuse into the interface between the ceramic and polymeric phases, potentially creating microcracks on the surface and further increasing the surface roughness of Cerasmart (Group S) compared to the other tested materials [38]. Lava Plus zirconia (Group Z) is a fully stabilized zirconia with a 5% yttria content, and it showed the lowest surface roughness value. The addition of yttria to the zirconia structure stabilized the crystal structure and made it more resistant to changes in the surface roughness even under different conditions, such as thermal cycling.

In this study, the contact angles for all materials tended to decrease after thermocycling. After thermocycling, the resin matrix of the CAD/CAM ceramic was expected to be more hydrophilic [39] (decreased WCA) due to water sorption during thermocycling. Sturz et al. [40] suggested that changes in the WCA may be attributed to the chemical composition of the material, particularly differences in the matrix composition and filler fraction, as well as potential surface inhomogeneities caused by the fillers.

In terms of biocompatibility, the ceramic restorations are in prolonged contact with oral soft tissues, especially with the marginal keratinized gingiva. Thus, biocompatibility is a critical characteristic that should be carefully evaluated prior to clinical use. In vivo, the formation of a mucosal seal around the prosthetic restoration is critical for success, and this process is highly dependent on the attachment of HGFs in the connective tissues to the substrate surface [41]. For cell viability, as revealed by the LIVE & DEAD assay, most cells remained viable on all surfaces, suggesting that all samples had excellent biocompatibilities toward HGFs. Except for the Lava plus zirconia (Group Z), the cell viability decreased with increasing aging time for all materials, as shown by the results of the CCK-8 assay. The vital reaction to dental restoration is affected by the surface topography and surface physiochemistry of the material used. The surface features of a material have a significant impact on the way cells interact with it. First, the surface wettability of a biomaterial, as determined with contact angle measurements, has been shown to have an impact on protein adsorption onto the surface of the material. Generally, a hydrophilic surface inhibits cell adhesion by preventing protein adsorption, while a hydrophobic surface promotes cell adhesion by inducing protein adsorption. Second, surface roughness has a significant impact on protein adsorption and subsequent cell behavior [42,43]. As shown by the results of the cell viability analysis, the present study revealed that surface roughness and wettability had significant effects on the initial stages of host tissue-biomaterial integration, and particularly on cell adhesion and proliferation. The impact of surface roughness on the attachment and proliferation of HGFs is a subject of debate, with some studies suggesting that smooth surfaces are more beneficial, while others report the opposite [44,45,46,47].

In this study, all of the materials exhibited increased surface roughness after thermal cycling aging, which resulted in reduced performance of the HGFs in terms of adhesion, proliferation, and migration. Therefore, based on the results of this study, it can be inferred that HGFs exhibit higher proliferation rates on smooth surfaces. Unlike other ceramic materials, zirconia showed a slight increase in cell viability in the aged group compared to the control group. This was assumed to arise because the surface of zirconia was very smooth, with a very low R_a_ value for surface roughness, even in the aged group, which may have led to a higher fibroblast survival rate [48]. In our study, Celtra Duo (Group C) exhibited a significantly lower cell viability than the other materials. It can be assumed that the absence of a crystallization process in the zirconia-reinforced lithium silicate (ZLS) may have resulted in an unstable crystalline structure of the material, which negatively impacted the cell viability [49]. Additionally, cytoskeletal organization provides the fundamental framework of the cell and maintains the cell shape, promotes cell maturation, and facilitates cell migration [50]. In the present study, it was observed that cell spreading of the HGFs was faster on smoother surfaces in the control group compared to rougher surfaces in the aged group, as shown by the F-actin intensities. Previous studies suggested that cells on smooth surfaces underwent spreading and developed robust actin cytoskeletons to achieve mechanical stability on smooth surfaces, whereas on rough surfaces, the topographical features of the surface were used by the cells for stabilization [51,52]. However, in this study, there were no differences in the cell areas of the control group and aged group after 24 h of seeding. The actin binding protein Talin1 is closely related to cell adhesion and is the key regulator of focal adhesion. Talin1 interacts with actin to form a connection with integrin, the main adhesion protein in cells, which helps cells attach to their surrounding matrix and spread [53]. Vinculin is involved in the formation of intercellular adhesions, cell spreading and movement, and regulates the structural stability of cells [54]. Talin1 and Vinculin proteins are important in the formation of cell adhesions, which are essential for the adhesion and proliferation of HGF on material surfaces. Consequently, Talin1 and Vinculin proteins are vital in the establishment of secure soft tissue seals around restoration materials. In Celtra Duo (Group C) and Vita Enamic (Group E), the fluorescence intensity of talin1 and vinculin proteins per cell was higher in aged samples, indicating stronger expression, whereas in the other groups, weaker signals were observed in aged samples. This suggested that various factors influenced cell adhesion, and based on these results, it can be inferred that this is attributable to changes in the chemical composition or surface topology after aging rather than the surface roughness. It is widely recognized that cell adhesion and growth on a material surface are affected by factors such as the chemical composition, roughness, and topography.

Considering the physical property perspective, the Lava plus zirconia biomaterial exhibited a surface hardness and elastic modulus that were more than double those of glass ceramics such as IPS e.max CAD and hybrid CAD/CAM ceramics such as Celtra Duo, Vita Enamic, and Cerasmart. This characteristic enables utilization of zirconia in a wide range of applications in dentistry, including long bridges and implant frameworks, where it effectively replaces the traditional metal materials. On the other hand, CAD/CAM zirconia blocks are predominantly made of partially sintered zirconia, as seen for the material used in this study (Lava plus zirconia), because fully sintered zirconia blocks are too strong and difficult to machine. However, partially sintered zirconia undergoes significant shrinkage during the sintering process, which can affect the accuracy of the final restoration. Hybrid CAD/CAM ceramics offer elastic moduli and surface hardnesses similar to those of natural teeth, and they exhibit excellent milling capabilities that enable faster fabrication of restorations and cause less wear on the milling equipment than zirconia. Since they already possess the desired final strengths, there is no need for additional sintering after milling, which eliminates inaccuracies due to shrinkage. Moreover, the time required for digital scanning, milling, polishing, and bonding is shorter, and the fabrication process is simpler than that for zirconia, which is advantageous [55,56]. Finally, we confirmed in vitro that the ceramic material used in this experiment was not cytotoxic. This demonstrated that the ceramic material used in this study shows promise in improving the esthetic outcomes of dental implants and in tightening the peri-implant junction.

From a biological perspective, the ceramic materials used in this study demonstrated that the HGFs prefer smooth surfaces, which underscores the recommendation for surface grinding and polishing based on faster cellular responses. In the case of implant placements, the marginal gingiva is injured, which initiates a regenerative process. Therefore, the HGFs migrate from the wound edges to the wound area, generate a collagen matrix and facilitate attachment of the keratinocytes. By modulating the surface characteristics, such as by polishing or grinding the implant material, tissue adhesion can be finely regulated, which enables the induction of tissue regeneration at the appropriate time.

The limitations of this study include the aging procedures. This artificial aging technique predicts the long-term degradation caused by aging the ceramic. However, in the oral environment, dental restorations are exposed to different stimuli, including mechanical aging, thermocycling, and chemical aging. Therefore, to reflect the clinical situation, more studies are required to assess the changes occurring in the physical properties, surface topography and cellular properties of dental ceramics under various aging conditions. Due to the in vitro nature of this study, further studies will be needed to define the technical protocols for the surface ceramic treatments and to understand the biological and mechanical advantages of these procedures. Furthermore, our study was limited to analyzing the expression patterns of cell adhesion proteins, and further research is needed to investigate the gene expression levels in more detail. Additionally, further research is needed on bacterial adhesion to CAD/CAM ceramics to reduce the risk of periprosthetic infections, as bacterial adhesion and biofilm formation on prosthetic surfaces are key steps in the pathogenesis of these infections.

## 5. Conclusions

The results of this study demonstrated that

i.The accelerated aging procedure induced changes in the physical properties of the CAD/CAM materials. The nanoindentation hardnesses and Young’s moduli were decreased after thermocycling aging. Lithium disilicate and hybrid CAD/CAM restorative ceramics tend to be lower than those of highly translucent zirconia before and after aging.ii.Cell viability and proliferation of the material decreased with aging except for the highly translucent zirconia materials. Significant differences were observed in the viabilities of the HGFs on the CAD/CAM materials. Zirconia-reinforced lithium silicate exhibited significantly lower cell viability than the other materials.iii.The surface roughness increased with aging for all groups, and the resin nanoceramic showed the highest roughness, while the highly translucent zirconia exhibited the smoothest surface among the materials. After aging, changes were observed in the surface microstructures, compositions, and hydrophilicities.iv.Cell adhesion and growth on a material surface are influenced by various factors, such as the surface chemical composition, hydrophilicity, roughness, and topography.

## Figures and Tables

**Figure 1 pharmaceutics-15-02122-f001:**
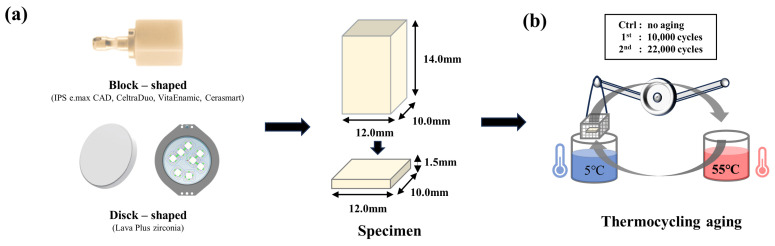
Standardization of a CAD/CAM specimen, Thermocycling aging, Nanoindentation hardness, Young’s modulus. (**a**) Block-shaped CAD/CAM restorative materials (IPS e.max CAD, Celtra Duo, Vita Enamic, Cerasmart) (Up), Disk- shaped CAM/CAM restorative material (Lava^TM^ Plus Zirconia) (Down), Design for milling in block form and fabrication of plate-shaped specimen (12.0 mm width × 10.0 mm length × 1.5 mm height) (Right). (**b**) Schematic diagram of Thermal aging.

**Figure 2 pharmaceutics-15-02122-f002:**
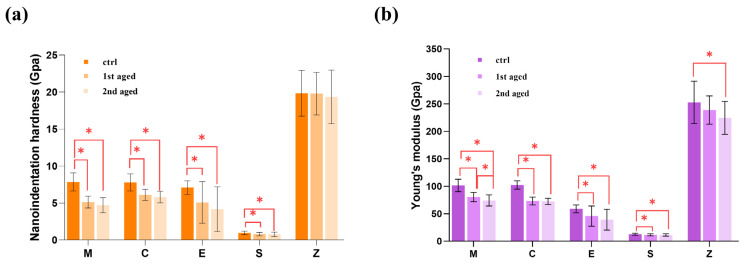
(**a**) Means and standard deviations for the nanoindentation hardness and (**b**) Means and standard deviations for the Young’s modulus. * denotes a significant difference at *p* < 0.05.

**Figure 3 pharmaceutics-15-02122-f003:**
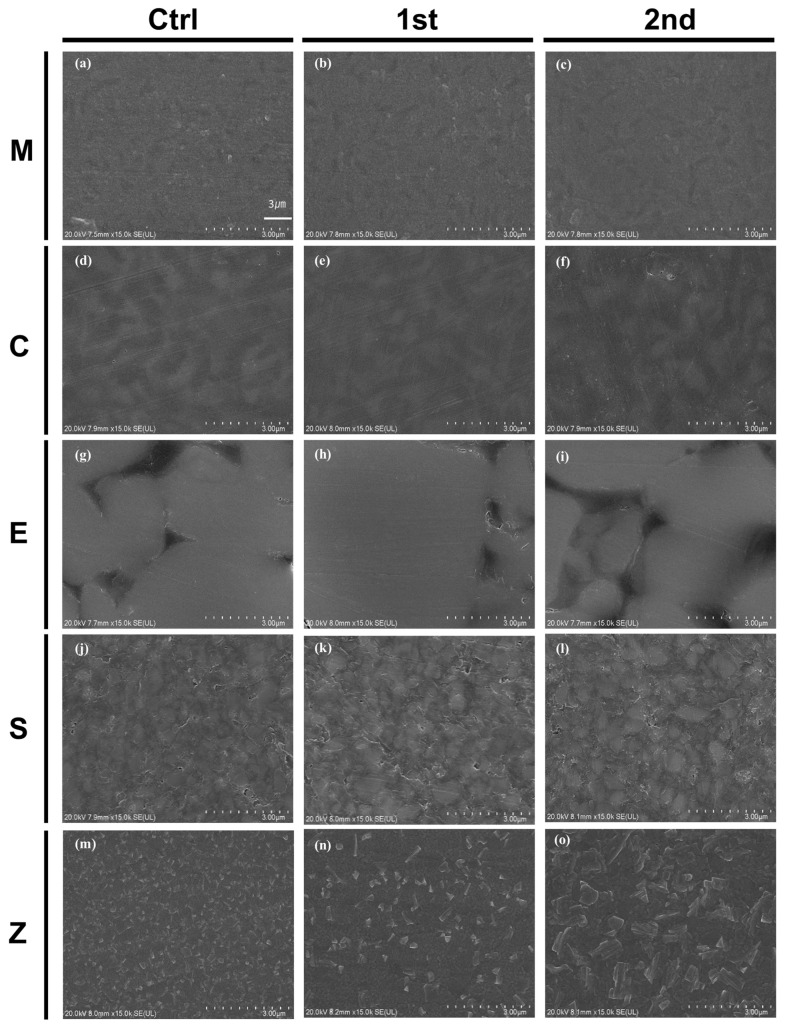
Field-emission scanning electron microscopy images of the surface topographies for all groups; (**a**) MC, (**b**) MAF, (**c**) MAS, (**d**) CC, (**e**) CAF, (**f**) CAS, (**g**) EC, (**h**) EAF, (**i**) EAS, (**j**) SC, (**k**) SAF, (**l**) SAS, (**m**) ZC, (**n**) ZAF, and (**o**) ZAS, (scale bar 3 μm).

**Figure 4 pharmaceutics-15-02122-f004:**
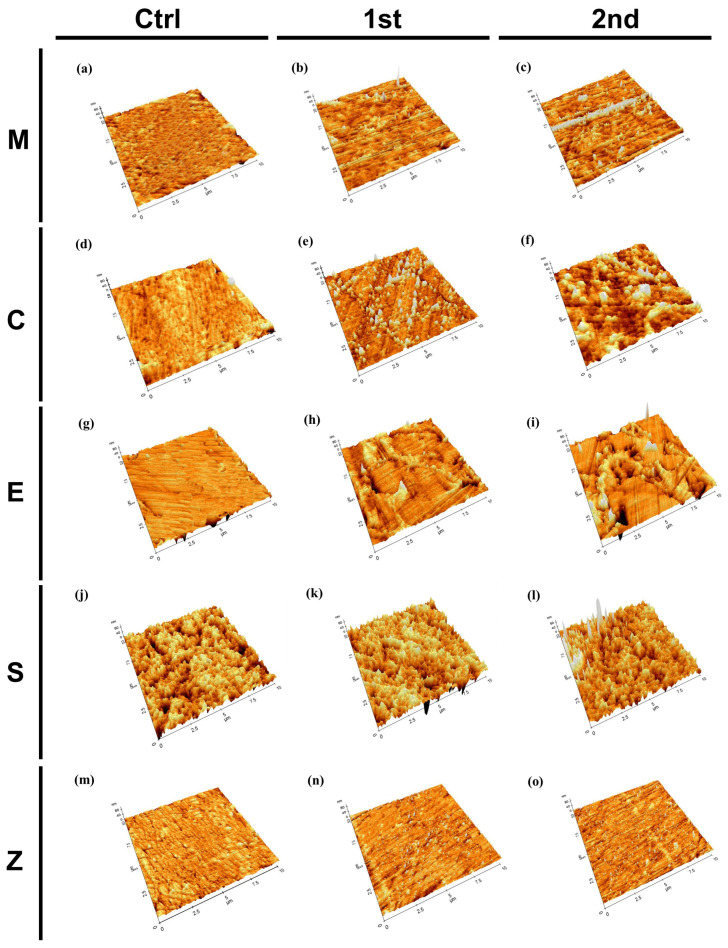
Atomic force microscopy images for all groups; (**a**) MC, (**b**) MAF, (**c**) MAS, (**d**) CC, (**e**) CAF, (**f**) CAS, (**g**) EC, (**h**) EAF, (**i**) EAS, (**j**) SC, (**k**) SAF, (**l**) SAS, (**m**) ZC, (**n**) ZAF, and (**o**) ZAS.

**Figure 5 pharmaceutics-15-02122-f005:**
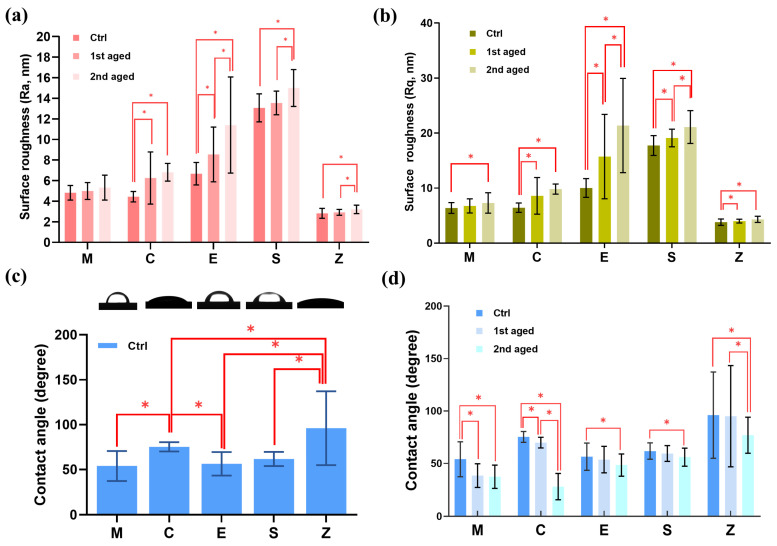
(**a**) Means and standard deviations for R_a_ (nm), (**b**) Means and standard deviations for R_q_ (nm). (**c**) Means and standard deviation and images for all control group of contact angle and (**d**) Means and standard deviations for the contact angles. * denotes a significant difference at *p* < 0.05.

**Figure 6 pharmaceutics-15-02122-f006:**
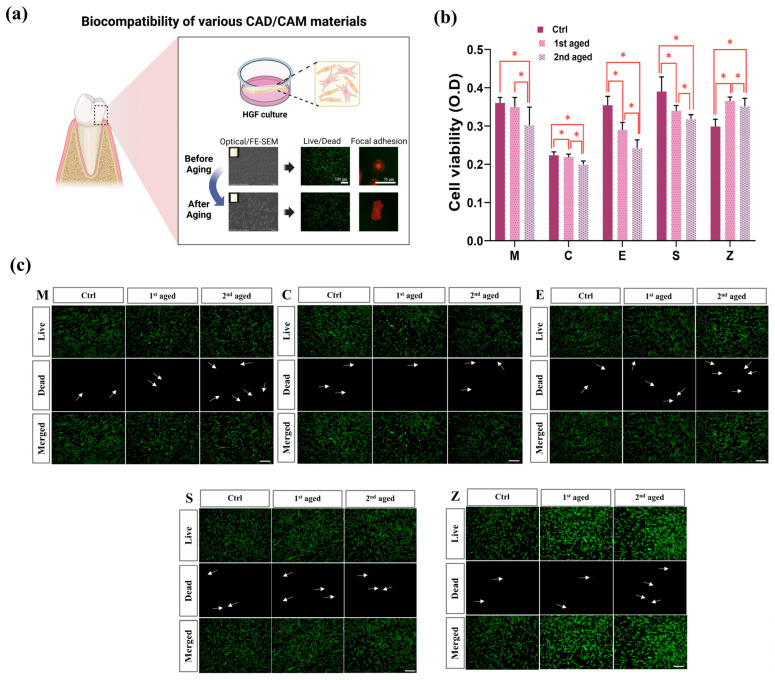
Cell viability analysis; (**a**) Schematic depicting the effects of aging on biocompatibility analysis of various CAD/CAM restorative materials. (**b**) CCK-8 assay performed on HGF cells cultured in all specimens for 24 h. Data are presented as mean ± SEM (n = 9). Statistical significance was determined using one-way ANOVA. * denotes a significant difference at *p* < 0.05. (**c**) Live/dead assay performed on HGF cells cultured in all specimens for 24 h. Live cells were stained green with calcein-AM and dead cells were stained red with ethidium homodimer-1. white arrow indicating dead cells. Data are presented as mean ± SEM (scale bar, 100 μm).

**Figure 7 pharmaceutics-15-02122-f007:**
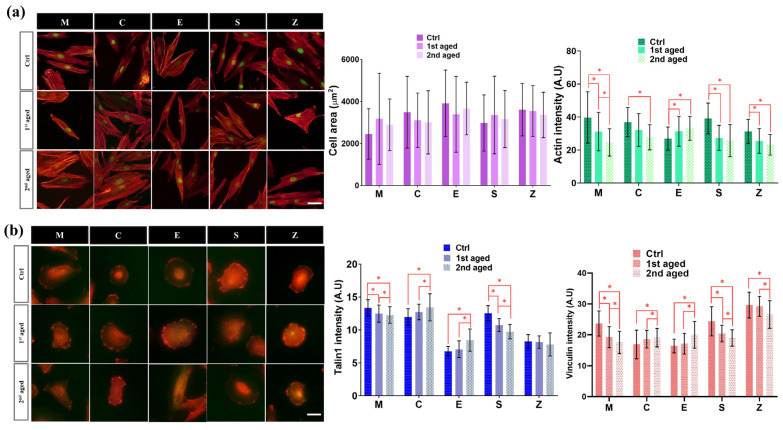
Immunofluorescence staining for focal adhesion and cell spreading; (**a**) Cell spreading. Representative immunofluorescence microscopy images of cytoskeletal organization (F-actin filament) stained with phalloidin (red), and nuclei stained with SYTOX™ (green) of HGF cells cultured for 24 h in all groups. The merged images are shown (scale bar, 50 μm). Graph shows quantification of cell spreading area and intensity using Image J software. Data are presented as mean ± SEM (n = 30). Statistical significance was determined using one-way ANOVA. * denotes a significant difference at *p* < 0.05. (**b**) Focal adhesion. Representative immunofluorescence microscopy mages with the focal adhesion markers indicating Vinculin (red) and Talin 1 (green) for HGF cells cultured for 4 h in all groups. The merged images are shown (scale bar, 50 μm). Graph shows quantification of focal adhesion intensity using Image J software. Data are presented as mean ± SEM (n = 30). Statistical significance was determined using one-way ANOVA. * denotes a significant difference at *p* < 0.05.

**Table 1 pharmaceutics-15-02122-t001:** Manufacturers and compositions of the materials used in the present study.

Product Name	Manufacturer	Shade (Lot No.)	Composition
IPS e.max CAD	Ivoclar Vivadent	2M2-HT (Y00999, Y26950)	0.2–2 μm lithium disilicate glass-ceramic (LS2, lithium disilicate)
Celtra Duo	Dentsply Sirona	A2-HT (16006746, 16006750)	silica-based glass matrix with 10% dissolved zirconia (ZLS, zirconia reinforced lithium silicate)
Vita Enamic	Vita Zahnfabrik	A2-HT (78540, 78880)	86 wt% mass percentage of inorganic ceramic network with 14 wt% of the organic polymer network infiltrated (PICN, polymer-infiltrated ceramic network)
Cerasmart	GC Corporation	A2-HT (1910101)	Polymer matrix (Bis-MEPP, UDMA, DMA) with 71 wt% silica and barium glass nanoparticles (RNC, resin nanoceramic)
Lava^TM^ Plus Zirconia	3 M ESPE	HT (260614)	Yttrium-stabilized zirconium dioxide crystals (Zirconia, high translucency zirconia)

Bis-MEPP: 2,2-bis (4-methyacryloxypolyethoxyphenyl) propane, UDMA: urethane dimethacrylate, DMA: dodecyl dimethacrylate.

**Table 2 pharmaceutics-15-02122-t002:** Abbreviations for the groups of tested materials.

Product Name	Material Code		Groups	
		Control	1st Aged	2nd Aged
IPS e.max CAD	M	MC	MAF	MAS
Celtra Duo	C	CC	CAF	CAS
Vita Enamic	E	EC	EAF	EAS
Cerasmart	S	SC	SAF	SAS
Lava^TM^ plus zirconia	Z	ZC	ZAF	ZAS

M: IPS e.max CAD, C: Celtra Duo, E: Vita Enamic, S: Cerasmart, Z: LavaTM Plus Zirconia.

**Table 3 pharmaceutics-15-02122-t003:** EDS elemental analyses for materials.

	Elemental Analyses (wt%)									
Materials	Group	C	O	Al	Si	P	K	Zr	Na	Ba
M	MC	5.90 ± 0.51 ^a^	54.10 ± 0.28 ^a^	0.81 ± 0.03 ^a^	33.70 ± 0.30 ^a^	1.51 ± 0.05 ^a^	3.98 ± 0.05 ^a^	-	-	-
	MAF	4.11 ± 0.57 ^b^	55.02 ± 0.33 ^b^	0.85 ± 0.03 ^b^	34.59 ± 0.34 ^b^	1.51 ± 0.09 ^a^	3.93 ± 0.07 ^b^	-	-	-
	MAS	3.78 ± 0.68 ^b^	56.62 ± 3.83 ^c^	0.81 ± 0.08 ^a^	33.66 ± 3.47 ^a^	1.51 ± 0.20 ^a^	3.62 ± 0.64 ^c^	-	-	-
C	CC	5.63 ± 0.62 ^a^	45.99 ± 0.49 ^a^	1.04 ± 0.03 ^a^	27.52 ± 0.22 ^a^	1.53 ± 0.07 ^a^	1.67 ± 0.03 ^a^	16.33 ± 0.35 ^a^	-	-
	CAF	4.32 ±0.93 ^b^	45.59 ± 0.57 ^b^	1.07 ± 0.03 ^b^	28.45 ± 0.42 ^b^	1.59 ± 0.10 ^b^	1.66 ± 0.06 ^a^	17.31 ± 0.32 ^b^	-	-
	CAS	7.30 ± 0.65 ^c^	52.16 ± 0.42 ^c^	1.11 ± 0.05 ^c^	27.65 ± 0.32 ^a^	2.52 ± 0.10 ^c^	1.58 ± 0.05 ^b^	7.69 ± 0.47 ^c^	-	-
E	EC	19.80 ± 1.14 ^a^	38.80 ± 0.52 ^a^	9.42 ± 0.24 ^a^	22.87 ± 0.63 ^a^	-	-	-	4.19 ± 0.17 ^a^	-
	EAF	18.97 ± 0.96 ^b^	38.89 ± 0.34 ^a^	9.58± 0.16 ^b^	23.37 ± 0.47 ^b^	-	-	-	4.31 ± 0.08 ^b^	-
	EAS	25.19 ± 4.65 ^c^	42.70 ± 2.53 ^b^	7.40 ± 1.36 ^c^	23.37 ± 0.47 ^b^	-	-	-	3.24 ± 2.16 ^c^	-
S	SC	25.45 ± 0.32 ^a^	33.05 ± 0.40 ^a^	3.32 ± 0.10 ^a^	19.61 ± 0.41 ^a^	-	-	-	-	18.57 ± 0.18 ^a^
	SAF	25.92 ± 0.45 ^b^	32.90 ± 0.78 ^a^	3.29 ± 0.07 ^b^	19.73 ± 0.52 ^a^	-	-	-	-	18.17 ± 0.60 ^b^
	SAS	26.17 ± 0.41 ^a^	32.97 ± 0.88 ^a^	3.25 ± 0.05 ^a^	19.46 ± 0.61 ^b^	-	-	-	-	18.15 ± 0.61 ^a^
Z	ZC	6.38 ± 0.32 ^a^	25.29 ± 0.16 ^a^	3.03 ± 0.15 ^a^	-	-	-	65.29 ± 0.16 ^a^	-	-
	ZAF	7.05 ± 0.61 ^b^	25.03 ± 0.28 ^b^	3.01 ± 0.07 ^a, b^	-	-	-	64.90 ± 0.39 ^b^	-	-
	ZAS	8.67 ± 0.76 ^c^	25.38 ± 0.28 ^a^	2.88 ± 0.25 ^b^	-	-	-	63.06 ± 0.55 ^c^	-	-

Different superscripted letters in each vertical column indicate significant differences (*p* < 0.05).

## Data Availability

The data that support the findings of this study are available from the corresponding author upon reasonable.

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
