# Peer review of "The Effects of Thermocycling on the Physical Properties and Biocompatibilities of Various CAD/CAM Restorative Materials"

_pharmaceutics, 2023, doi:10.3390/pharmaceutics15082122_

Round 1

Reviewer 1 Report

The article “The Effects of Thermocycling on the Physical Properties and Biocompatibilities of Various CAD/CAM Restorative Materials” investigated the physical and surface properties as well as the biocompatibilities of materials under thermal cycling aging.

The article is interesting, lots of variables have been investigated, and the reviewer suggests some minor changes to improve the manuscript further.

Line 38:

Cell adhesion and were influenced

and what?

Lines 46-7:

“Ceramics exhibit high wear resistance, aesthetics, and excellent color stability, as well as a high degree of biocompatibility. “

Please support this sentence with references.

The reviewer suggests for aesthetics and color stability the following paper:

Paolone, G., Mandurino, M., De Palma, F., Mazzitelli, C., Scotti, N., Breschi, L., Gherlone, E., Cantatore, G., & Vichi, A. (2023). Color Stability of Polymer-Based Composite CAD/CAM Blocks: A Systematic Review. Polymers15(2), 464. https://doi.org/10.3390/polym15020464

Lines 50-2:

The authors could add the above-mentioned reference also to support this sentence.

Line 60:

Please rephrase “and the bur lives are longer. ”

Line 74:

please remove “On the other hand, ”

The authors should outline that the classification of resin-based CAD/CAM restorative materials is still not well defined among the scientific community. Some authors define these materials (Cerasmart) such as composites, other such as resin nano-ceramic, etc.

It should be outlined that they are all resin-based materials with different types of fillers (therefore they are composites).

Enamic is a completely different material (PICN) since it is a ceramic network infiltrated with ceramic.

So they are hybrid but the reviewer thinks that it should be outlined that the distinction is between PICN (which is only Enamic) and Composite blocks.

This is declared in lines 421-424 but should be better defined in the introduction.

Lines 531-3

The authors could find other papers supporting the following sentence:

“Moreover, the time required for digital scanning, milling, polishing, and bonding is shorter, and the fabrication process is simpler than that for zirconia, which is advantageous. ”

The authors could change the Conclusion’s style in a bullet point one. The authors have analyzed many variables, and a bullet/sentence per variable could help the reader to quickly understand which results were achieved.

little rephrase of a sentence

Reviewer 2 Report

Please see the questions and comments in the attached document.

Reviewer 3 Report

Dear author,

Congratulation for this work a lot of results in a very good paper!!

Please remove i), ii) and iii) from The null hypothesis. Please write normal

Figura 1 and b should be  Schema of.... /Schematic.....Please split from c and d

Please increase the font in Figure 4. Should be the same with font from article or close.

Fig 4 and 5. Please increase font of size inside of image

Figure 6. increase size of graphs

Very good paper! Would be nice all to be the same.

good
